# Word Recognition with a Cochlear Implant in Relation to Prediction and Electrode Position

**DOI:** 10.3390/jcm13010183

**Published:** 2023-12-28

**Authors:** Annett Franke-Trieger, Susen Lailach, Joshua Shetty, Katrin Murrmann, Thomas Zahnert, Marcus Neudert

**Affiliations:** Department of Otorhinolaryngology, Head and Neck Surgery, Faculty of Medicine Carl Gustav Carus, Technische Universität Dresden, Fetscherstraße 74, 01307 Dresden, Germanythomas.zahnert@ukdd.de (T.Z.);

**Keywords:** cochlear implant, WRS prediction, insertion depth, word recognition score, lateral wall, digital volume tomography

## Abstract

Background: the word recognition score (WRS) achieved with cochlear implants (CIs) varies widely. To account for this, a predictive model was developed based on patients’ age and their pre-operative WRS. This retrospective study aimed to find out whether the insertion depth of the nucleus lateral-wall electrode arrays contributes to the deviation of the CI-achieved WRS from the predicted WRS. Materials and methods: patients with a pre-operative maximum WRS > 0 or a pure-tone audiogram ≥80 dB were included. The insertion depth was determined via digital volume tomography. Results: fifty-three patients met the inclusion criteria. The median WRS achieved with the CI was 70%. The comparison of pre- and post-operative scores achieved with a hearing aid and a CI respectively in the aided condition showed a median improvement of 65 percentage points (pp). A total of 90% of the patients improved by at least 20 pp. The majority of patients reached or exceeded the prediction, with a median absolute error of 11 pp. No significant correlation was found between the deviation from the predicted WRS and the insertion depth. Conclusions: our data support a previously published model for the prediction of the WRS after cochlear implantation. For the lateral-wall electrode arrays evaluated, the insertion depth did not influence the WRS with a CI.

## 1. Introduction

Cochlear implantation is an established treatment option for patients with hearing loss for which hearing aids (HAs) or other less invasive options have failed to restore speech perception to a sufficient degree [1]. The vast majority of cochlear implant (CI) recipients show improved word recognition scores (WRSs) if the pre-operative-aided scores at a conversational level of 65 dB (WRS_65_(HA)) and post-operative scores with a CI (WRS_65_(CI)) are compared [2,3,4,5,6,7,8,9]. A number of pre-, intra- and post-operatively assessed outcome-predicting factors have been identified [2,3,4,5,6,7]. Blamey et al. found five intrinsic factors that had an impact on the post-operative word recognition score: the duration and age of onset of severe-to-profound hearing loss, age at the time of surgery, aetiology, and implant experience. Additionally, Holden et al. [3] identified extrinsic factors, such as scalar location, insertion depth, array insertion depth, angular position of the basal electrode’s contact, and wrapping factor as affecting word recognition.

Recent studies [10,11,12] have revealed the variability in electrode array positioning. This is partially due to differences in cochlear size, scalar shifts, and different electrode designs [13,14,15]), causing different electrophysiological findings [16] and different intracochlear trajectories of the electrode array [17]. In measurements in vivo, the insertion depth angle (AID) was found to vary by up to 300° for certain electrode arrays [10].

Placement shift due to scalar shift did not result in coherent findings with respect to speech comprehension. Liebscher et al. [12] did not find measurable differences in the WRS, whereas Aschendorff et al. [18] reported a detrimental effect of dislocation of up to 10 percentage points (pp) for the WRS of patients with scalar dislocations. Furthermore, the surgeon represents another source of variability in the electrode’s position; this might be intended for certain techniques, such as the pull-back technique [19,20], or be due to the placement of the electrode array in the markings specified by the implant manufacturers, which can cause variability in the distance between the first electrode’s contact and the round window, depending on the electrode array.

The position of the electrode array does affect electrophysiological measures, such as electrically evoked compound action potential, ECAP [11]. Therefore, the question arises of whether the electrode position has an influence on a CI’s performance.

However, for the comparison of both perimodiolar and lateral-wall electrode arrays and the influence of insertion depth, no consistent results have been shown; this may be due in part to the inhomogeneity of the patient groups analysed.

To account for the variability in audiological outcomes, significant efforts in recent years have focused on developing valid and reliable predictive models. In recent studies, Hoppe et al. proposed [5] and validated [9] a prediction model with a comparatively low prediction error (mean absolute error, MAE) of 11.5 pp [7,9].
(1)$${WRS}_{65}\left(CI\right)\left[\%\right]=\frac{100}{1+e^{-\left(\beta_{0}+\beta_{1}\cdot {WRS}_{max}+\beta_{2}\cdot age+\beta_{3}\cdot {WRS}_{65}\left(HA\right)\right)}}$$
with β_0_ = 0.84 ± 0.18 β_1_ = 0.012 ± 0.0015 1/% β_2_ = −0.0094 ± 0.0025 1/years β_3_ = 0.0059 ± 0.0026 1/%.

The model is based on pre-operative audiometric measures only: the maximum word recognition score (WRS_max_), the WRS_65_(HA), and the recipient’s age at implantation. This outcome prediction model can facilitate the pre-operative counselling of HA users [5,21]. Furthermore, Hoppe et al. found that the WRS_max_ is a highly reliable minimum predictor [4]. Both of these measures can be used within post-operative CI aftercare to set an expectation value (and post-operative objective) for WRS_65_(CI). This predicted WRS_65_(CI) can be used to monitor and reference the patient’s progress and, if appropriate, to reallocate clinical resources to improve outcomes [9]. In a recent study [22], the model was applied to investigate the systematic differences between CI recipients’ reaching or missing the predicted WRS_65_(CI). For this purpose, Dziemba et al. [22] expanded the exponential term in Equation (1) with additional factors representing post-operative audibility and loudness growths. They found that there are systematic differences between poor- and well-performing subjects; these differences are basically due to CI system fitting.

To our knowledge, this model has not yet been used to investigate contributing factors such as electrode positioning [3].

In the evaluation of the electrode array position, a distinction must be clear between lateral-wall and perimodiolar electrode arrays. While Liebscher et al. [12] determined the relationships between surgical technique, speech perception, electrophysiological parameters, and scalar translocations for perimodiolar electrode arrays, no information exists yet on the influence of surgical insertion on outcomes when a lateral-wall electrode of the same implant generation is used. On one hand, prediction models can contribute to more precise patient counselling, and on the other, they can be used for quality assurance measures, since a precise therapy target can be defined. In cases of deviation from the prognosis, pre-operative parameters (anamnesis, aetiology, and anatomy), intra-operative factors (electrode array insertion), and post-operative aspects (fitting and rehabilitation strategy) have to be examined.

Consequently, this study aimed to answer the question of whether optimising intra-operative process quality (i.e., in this case, by optimising the insertion depth) can reduce the deviation from the predicted WRS. Furthermore, one must ask whether the insertion depth contributes to the variability in the deviation from the prediction. The relationship between angular insertion depth and cochlear size, as well as the influence of the surgeon, was investigated. By varying the insertion depth alone, the surgeon could potentially contribute to the variability in the outcome of cochlear implantation.

In this study, we analysed the WRS to determine the influence of electrode position (the angular insertion depth and the distance between the most basal electrode contact and the round window).

## 2. Materials and Methods

### 2.1. Subjects

We reviewed all adult patients who received a Cochlear™ Nucleus^®^ implant (Cochlear Ltd., Sydney, Australia) with lateral-wall electrode arrays (CI522 or CI622) at the University Hospital of Dresden between May 2015 and June 2021. The two implant types have identical lateral-wall electrode arrays and functions. The receiver/stimulator housings differ. The inclusion criteria for this study were: sensorineural origin of hearing loss, post-lingual onset of deafness, native German speaker, imaging of the cochlea without pathological findings or malformations, age at implantation ≥18 years, and regular visits to the rehabilitation centre for fitting, audiometric testing, and hearing therapy. The hearing loss for air conduction was determined as the mean value over the frequencies 0.5, 1, 2, and 4 kHz (PTA4). For hearing thresholds beyond the maximum possible presentation levels of the audiometers, a value of 120 dB_HL_ was assigned. With respect to pure tone and speech audiometry, only patients with WRS_max_ > 0% or PTA4 ≥ 80 dB_HL_ were included. Furthermore, only patients with correct intracochlear electrode positioning were included. This was verified using a digital volume tomography image.

This study was conducted in accordance with the Declaration of Helsinki (2013) on research involving human subjects and was approved by the local ethics committee (SR+BO-260052021). The study was also registered under DRKS00026741 with the German register of clinical studies.

### 2.2. Audiometric Measures

Speech audiometry was performed using the Freiburg monosyllabic word test. The pre-operative WRS was measured with headphones in the unaided condition. To identify WRS_max_, the presentation level was increased in steps of 10 dB until the maximum score achievable (WRS_max_) below the patient’s loudness discomfort level was reached [5]. The WRS in the aided condition, i.e., with hearing aids (WRS_65_(HA)) and with the cochlear implant (WRS_65_(CI)) was measured in an anechoic soundproof booth at a loudspeaker presentation level of 65 dB SPL, with the patient seated 1.0 m in front of the loudspeaker. The measurements were performed monaurally. If necessary, the contralateral ear was appropriately masked with wideband noise presented through the headphones (DT48; beyerdynamic GmbH & Co. KG, Heilbronn, Germany). Speech audiometry was performed with an AT900 or AT1000 clinical audiometer (Auritec GmbH, Hamburg, Germany). The WRS with the cochlear implant referred to the score measured twelve months after the first activation of the CI system. To calculate the prognoses of the WRS_65_(CI), Equation (1) was used. Significant differences between WRSs were determined according to their critical differences according to Winkler and Holube [23].

### 2.3. Imaging

The flat panel volume tomography (digital volume tomography, DVT) examinations were carried out on the first day after implantation using a Flat Panel Computer Tomograph 3D Accuitomo 80 (J. Morita MFG. CORP., Kyoto, Japan). The imaging was performed with a tube current of 8 mA and a tube voltage of 90 kV. The raw projection images were reconstructed using i-dixl software (version 2.8., J. Morita MFG. CORP. Kyoto, Japan), resulting in a voxel size of 125 µm.

### 2.4. Measurement of Cochlear Diameter and Electrode Position

The angle and length measurements were performed according to the consensus paper [24] using the cochlear view, which is defined as the plane through the basal turn and perpendicular to the modiolus. Figure 1 shows an example of this measurement. The zero-degree reference angle was chosen at the centre of the round window according to the consensus paper. To quantify the cochlear size, the diameter of the basal turn of the cochlear was measured. This diameter is illustrated with the line that starts at the centre of the round window and crosses the position of the helicotrema and the ends of the lateral wall on the opposite side, as shown in earlier studies [25]. The distance, *d,* between the round window and the most basal electrode contact, was measured as shown in Figure 2. *d* is a parameter that describes how deeply the surgeon inserted the electrode array into the cochlea.

### 2.5. Data Analysis

All analyses and figure creations were produced using OriginLab (version 2019, OriginLab software, Northampton, MA, USA). The correlation analysis was performed using Spearman’s rank correlation method.

## 3. Results

### 3.1. Study Cases

Of 312 cochlear implantations carried out in the study period, 53 cases (i.e., individual ears; 34 right, 19 left) were identified that met the inclusion criteria. In all cases, implantation was carried out via round window insertion or via the extended round window approach. The implanted device was the CI522 in 37 cases and the CI622 in 16 cases. The age of the patients at implantation ranged from 26 to 80 years (mean: 61.4 years). The mean hearing loss for air conduction using the PTA4 was 80 ± 15 dB_HL_. 

Figure 2 shows the relationship between the pre- and post-operative WRSs. The median score achieved with the CI was 70% with the first quartile at 60% and the third quartile at 80%, as shown in Figure 2. Comparing the pre- and post-operative scores achieved in the aided condition showed a median improvement of 65 pp. In all cases, 90% improved by at least 20 pp. With respect to the minimum prediction, 96% of the recipients reached or exceeded the WRS_max_ while 83% of the recipients significantly exceeded the pre-operative WRS_65_(HA) [23].

Figure 3 shows the distribution of differences between the measured and predicted WRSs (measured minus predicted). The differences range from −57 pp to +35 pp. The MAE was 11 pp. Three patients missed the predicted score by more than 20 pp.

### 3.2. Insertion Depth and Cochlear Size

Figure 4 shows the angular insertion depth as a function of the diameter (*A*) and of the distance, *d,* as defined in Figure 1. The diameter ranged from 8.05 mm to 10.34 mm. The median diameter was 8.96 mm. The distances, *d,* ranged from 1.5 mm to 8.3 mm. The median distance was 4.7 mm. The angular insertion depth ranged from 365° to 568°. The median angle was 460°.

A positive correlation was found between the distance, *d,* and the resulting angular insertion depth. A negative correlation was found between the cochlear diameter and angular insertion depth. The data show that the correlation between *d* and the angular insertion depth was stronger (*r* = 0.673, *p* < 0.0001) than the weak correlation between the angular insertion depth and cochlear diameter (*r* = 0.306, *p* = 0.0254).

### 3.3. Dependence of the WRS on the Electrode’s Position and Cochlear Size

Figure 5 shows the difference between the measured and predicted WRSs as a function of the distance, *d,* and the angular insertion depth. The correlation analyses showed no significant correlation between the deviation from the predicted WRS and the distance *d* (*r* = −0.256, *p* > 0.05) and the angular insertion depth (*r* = −0.185, *p* > 0.05).

## 4. Discussion

The extension of the CI indication to patients who still have a capacity for speech perception inevitably creates enormous demands on the quality of care. In addition to pre-operative selection and counselling based on current audiological performance with and without a hearing aid, knowledge of potential surgical influencing factors and electrode array characteristics potentially contribute to the best possible hearing result by modulating these factors as necessary.

This study showed that 83% (44/53) of patients had clinically significantly ([23]; see also Methods) better WRSs after cochlear implantation than before with conventional hearing aids. The median improvement was 65 pp, and 90% of the patients showed an improvement of at least 20 pp. This is consistent with the results of earlier studies that also analysed word recognition with CIs in patients with residual hearing [4,8,26]. WRS_65_(HA) alone is not suitable for predicting WRS_65_(CI) post-operatively. Regression models only explain up to 10 pp of the WRS_65_(CI) [5]. More than half of our patients had a pre-operative WRS_65_(HA) of 0% even though the WRS_max_ was larger than zero. This finding, i.e., that the WRS_max_ is not met by the WRS_65_(HA), is in accordance with the results of previous studies [4,5,8]. However, even this patient group was able to achieve a mean WRS_65_(CI) of 65%, with a range from 0% to 90%. The inclusion of additional pre-operative speech audiometry measures may help to improve outcome prediction in this subgroup of recipients [26]. In contrast to WRS_65_(HA), a stronger association of WRS_65_(CI) with the pre-operative WRS_max_ was shown. Other research groups have already been able to identify this correlation [4,8,26]. These results suggest that patients with a pre-operatively great difference between WRS_65_(HA) and WRS_max_ (speech perception gap) benefit from cochlear implantation [27]. Especially in patients with severe hearing loss, sufficient hearing aid fitting often fails, owing to technical limitations (feedback), the lack of acceptance of high sound levels, and a low dynamic range [28]. In our study, the WRS_65_(CI) was below the WRS_max_ in only two patients. With respect to the minimum prediction, 96% of the recipients reached or exceeded their WRS_max_.

The majority of patients achieved or exceeded the WRS_65_(CI) predicted according to Equation (1). Three patients missed the predicted score by more than 20 pp. The prediction model was thus also confirmed with our study. In the validation process of the prediction model, Hoppe et al. [9] determined an MAE of 11.5 pp in a patient group with a WRS_max_ above zero. Additionally, they reported that 14 out of 85 patients missed the predicted score by more than 20 pp. For all patients with WRS_max_ = 0%, they reported an MAE of 23 pp. In our study, in cases with WRS_max_ > 0 or PTA4 ≥ 80 dB HL, the MAE was 11 pp.

The modelling of prognosis prediction by Hoppe et al. was based on a group of patients fitted with a perimodiolar electrode array. Our investigations confirm that the model can also be applied to patients with lateral-wall electrode arrays. In the current literature, no significant difference in speech comprehension between perimodiolar and lateral-wall electrode arrays can be found, although the heterogeneous quality of the studies does not allow a conclusive evaluation [29,30]. Especially for perimodiolar electrode arrays, the optimisation of the electrode array position is aimed at improving surgical techniques, e.g., the pullback technique, to achieve the smallest possible distance between the electrode array and the modiolus [19,31]. The results of our study suggest that such procedures are probably not necessary for the CI522/CI 622 implants, as the electrode array position ultimately has no influence on the audiological outcome.

While the pre-operative WRS_max_ could be confirmed as a strong minimum predictor, the insertion depth had no influence on the post-operative WRS_65_(CI) in our study. The cochlear coverage could be influenced by the cochlear duct length (CDL) and the insertion depth of the electrode array. For CI systems with different available electrode lengths, the coverage is of course influenced by the chosen electrode’s length. For the CI622/CI522 implants, the cochlear coverage is determined basically only by the CDL and the distance, *d* (first electrode contact to the round window). Up to now, no information has become available on the extent to which *d*, which ultimately is determined by the surgeon, influences post-operative performance. According to the physician’s guide [32] provided by the implant company, the white markers, which are positioned 20 mm and 25 mm away from the apical tip of the electrode array, are currently used as a guide for insertion depth, and a maximum insertion depth of 25 mm is assumed. Deeper insertion was not considered necessary by the implant company, although no study data were presented to support this recommendation.

For other electrode manufacturers, especially those with different electrode lengths in their portfolio, the exact pre-operative planning of the electrode array position based on the CDL and the residual hearing was discussed [33]; however, this does not seem to be necessary for the CI622/CI 522 implants with normal cochlear anatomies. The influence of insertion depth on word recognition after implantation is frequently discussed in the current literature. While some authors have demonstrated better word recognition with deeper insertion in lateral-wall electrode arrays [34,35,36,37], this effect has been disputed by other research groups [38,39,40]. Some studies even showed a worse speech audiometric outcome with deeper insertion [38,41]. In most of these studies, all lateral-wall electrode arrays of all the available manufacturers were combined, so that no implant-specific recommendations could be derived from them. Other studies focused exclusively on implants from other manufacturers so the results cannot be applied to Nucleus implants, especially to the CI522/CI622 implants used in this study. Last but not least, the level of evidence of the current studies on the influence of insertion depth on audiological performance is currently not satisfactory [42]. Often, there is a lack of adequate consideration of additional known confounding factors and an adequate control group. The practice of switching off the apical electrode contacts to simulate a shortened insertion depth must also be critically questioned [40] since it is known that the number of active electrode contacts also contributes to word recognition.

Various hypotheses exist to explain the possible influence of insertion depth on post-operative word recognition. On the one hand, a greater insertion depth is considered to afford a better coverage of the spiral ganglia in the low-frequency range and a more physiological frequency assignment [40,43]. However, other authors presume a greater trauma for cochlear structures with deeper insertion [44]. In the case of shorter electrode arrays, individual authors have found a poorer outcome with deeper insertion, since the basal region is not sufficiently covered, owing to the greater distance between the first electrode contact and the round window [3,41]. This could not be confirmed in our study for the investigated electrode array of the CI622/CI 522 implants with an active length of 19.1 mm. Here, however, we should point out that these results cannot simply be transferred to electrode arrays from other companies. It should finally be observed that the debate regarding the ideal length of an electrode array and its ideal cochlear coverage, which has been going on for years (partly for reasons of marketing strategy) cannot at present be resolved.

The distance, *d,* from the first electrode contact to the round window, is the aspect of the insertion depth of the electrode array that can be determined and controlled by the surgeon himself. In our study, *d* was found to vary from 1.5 mm to 8.5 mm. When evaluating the scatter of *d* and insertion depth, the measurement error of the angle and length measurement based on the post-operative DVT must also be taken into account. In the literature, interrater differences of −0.5 to 0.5 mm for length measurements and 12° to 30° for angle measurements can be found [45].

In addition to the insertion depth, the aspect of structural preservation through atraumatic electrode insertion is currently under discussion. Therefore, studies are currently being performed to evaluate the influence of insertion speed and insertion force on the outcome of cochlear implantation. The preservation of residual hearing is primarily evaluated as a correlation of structural preservation. In recent years, electrocochleography has been implemented as a system for monitoring residual hearing in individual clinics [46,47,48,49,50]. Structural preservation as a function of insertion depth or residual hearing preservation was not assessed in our study. In the literature, the influence of insertion depth on residual hearing preservation is currently a topic of controversy. While some authors see the advantages of a lesser insertion depth, which is associated with less severe intracochlear trauma [44], other research groups have been able to demonstrate satisfactory residual hearing retention even with deeper insertion [51,52,53]. To summarise, at the moment is not clear how the WRS is affected by the factors discussed above. More studies are needed on the effect of these different factors on the WRS. To mention one example, Dalbert et al. [54] demonstrated better long-term speech understanding in patients with residual hearing than in the group of patients without residual hearing for patients with electrical stimulation alone. However, the significant positive effect in the study group was not seen until 18 months after CI activation.

Owing to the great heterogeneity of these studies, it is not currently possible to conduct a high-quality meta-analytical review of the relationship between insertion depth and speech comprehension. In a systematic review published in 2021 including seven studies with results of speech comprehension after one year, the effect of insertion depth could not be reliably assessed [42]. Because of the improvement in word recognition within the rehabilitation process, an assessment after less than 12 months does not seem to be very meaningful; however, after 12 months, stable speech comprehension can be assumed [3]. Interestingly, Büchner et al. [55] observed that the initially positive effect of a longer electrode array length diminished over the course of rehabilitation. The authors attributed this to cortical plasticity, which can compensate for any possible frequency mismatch present [55].

One limitation of our study is the lack of a systematic analysis of the fitting. Some studies have already shown the strong effect of fitting quality on the outcome of cochlear implantation. Thus, currently, high variability in audiological outcomes due to a less-than-optimum fitting is possible [22,56,57]. However, standardised quality indicators for the evaluation of fitting quality must be developed and analysed in further studies with the help of a prediction model. For example, by basing the adjustment on the electrode-specific ECAP or a categorical loudness scale, it should be possible to reduce the error caused by the adjustment.

It should also be mentioned that the sole outcome parameter was the WRS at 65 dB; speech perception in noise and subjective hearing perception, e.g., music hearing, were not assessed. In further studies, one might investigate to what extent the position of the electrode array affects these other outcome parameters since a frequency mismatch could possibly be more important here.

## 5. Conclusions

Our results support the previously published model for predicting outcomes after cochlear implantation. WRS_max_ plays a more important part than WRS_65_(HA), by allowing the prediction of the outcome of cochlear implantation. With the help of the prediction model, improved pre-operative counselling of patients on the expected outcome can be provided for patients with a pre-operative WRS_max_ greater than zero. For the implants used (CI622 and CI522) the insertion depth did not influence the post-operative outcome. The surgeon did not influence the outcome positively or negatively according to the distance (*d*) from the first electrode contact to the cochlear window within the observed range.

## Figures and Tables

**Figure 1 jcm-13-00183-f001:**
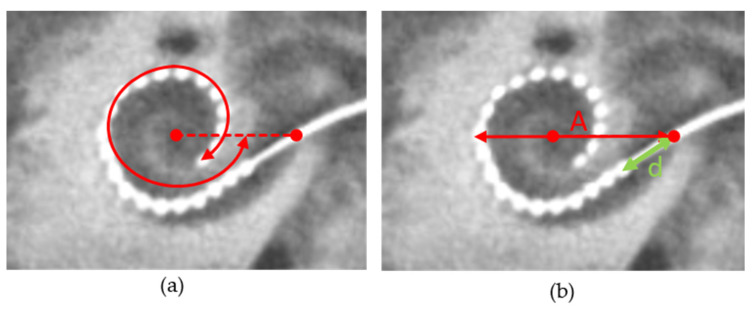
Cochlear view of the DVT image with the red dots indicating the position of the modiolus and the position of the round window. (**a**) Measurement of the insertion depth angle. (**b**) Measurement of the diameter of the cochlear basal turn (*A*) and the distance between the round window and the most basal electrode contact (*d*).

**Figure 2 jcm-13-00183-f002:**
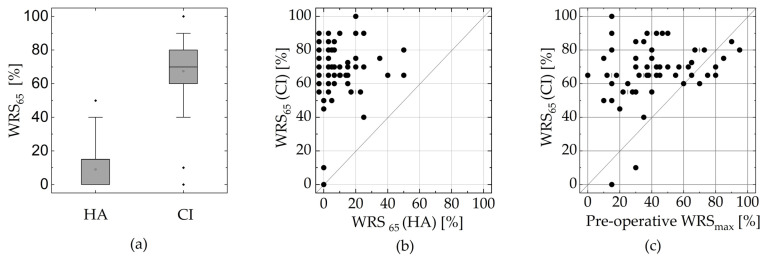
Relationship between pre- and post-operative audiometric measures. (**a**) Box plot comparing the pre-operative WRS_65_(HA) and post-operative WRS_65_(CI); the boxes show the quartiles and the whiskers show the 5th and 95th percentile; the median for HA lies on the lower edge of the box. (**b**) Scatter plot showing the same comparison. (**c**) Comparison between the pre-operative WRS_max_ and post-operative WRS_65_(CI). In (**b**,**c**), the overlapping points are shifted apart horizontally, with a small vertical line representing their actual position.

**Figure 3 jcm-13-00183-f003:**
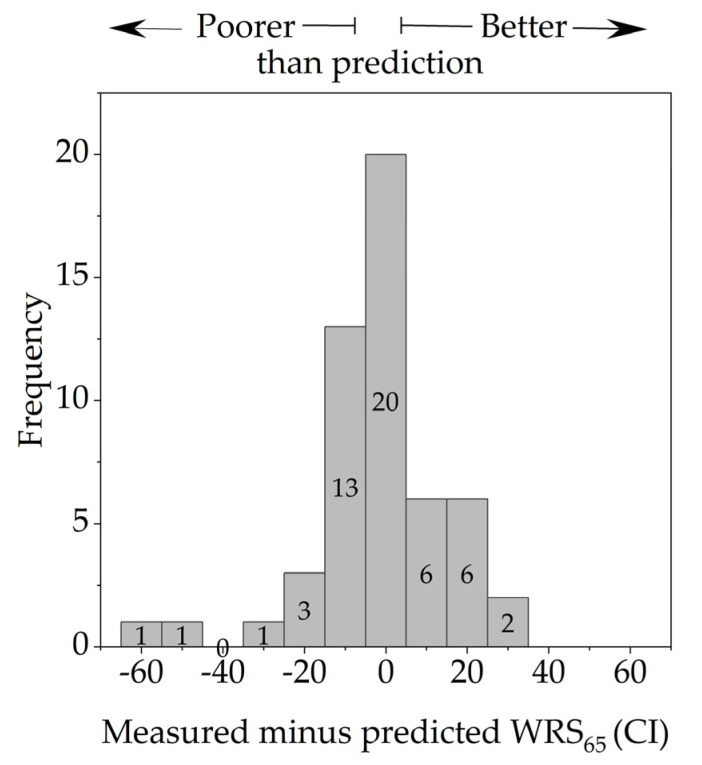
Distribution of differences (in percentage points) between the measured and predicted word recognition scores. Negative differences correspond to cases in which the measured scores were below the predictions.

**Figure 4 jcm-13-00183-f004:**
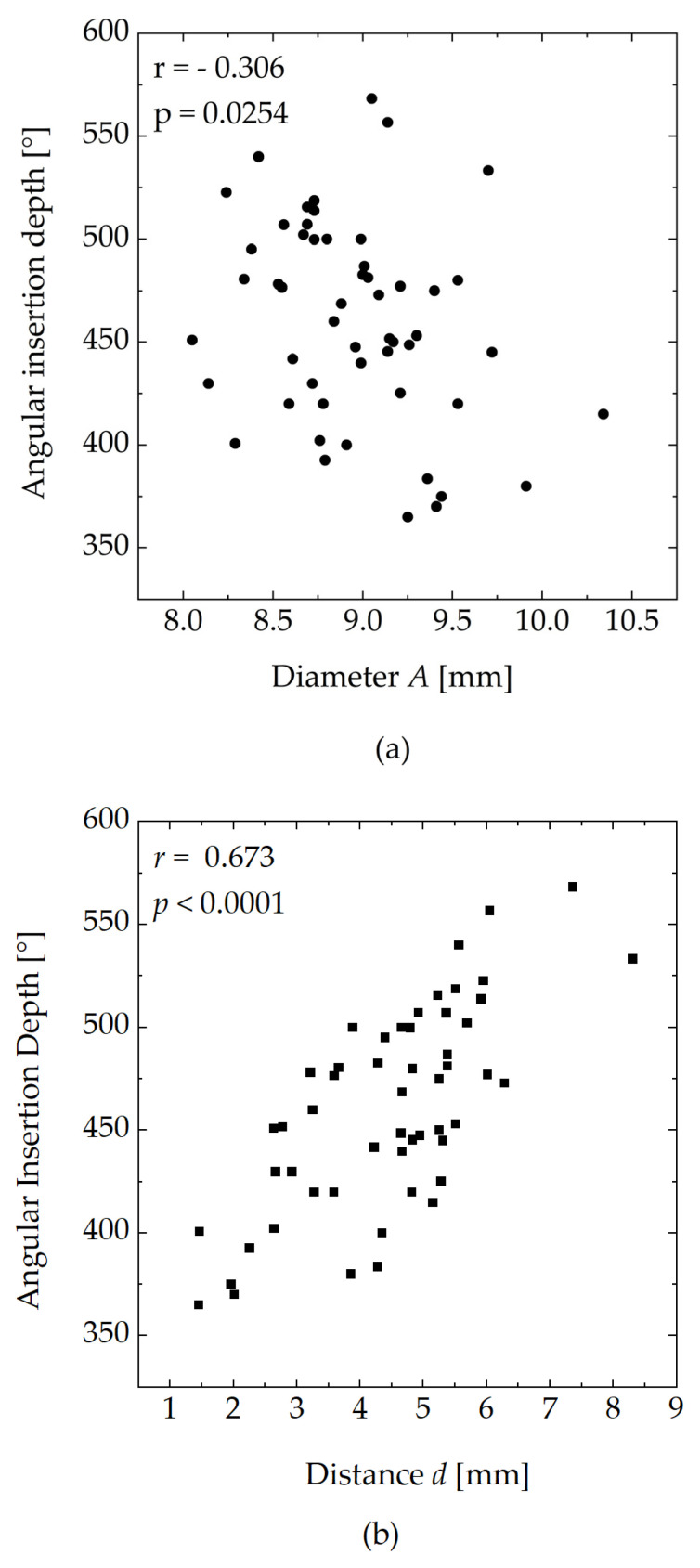
(**a**) Angular insertion depth as a function of the diameter of the basal turn *A*. (**b**) Angular insertion depth as a function of the distance, *d*, both defined in Figure 2. *r*, Spearman rank correlation coefficient.

**Figure 5 jcm-13-00183-f005:**
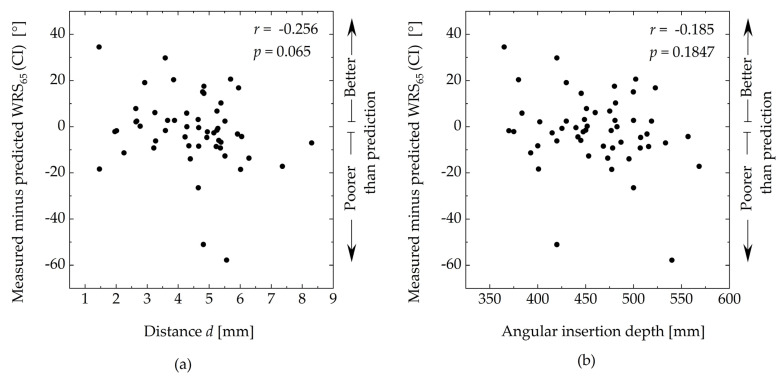
The difference between the measured and predicted word recognition scores as a function of (**a**) the distance, *d,* and (**b**) the angular insertion depth. *r*, Spearman rank correlation coefficient.

## Data Availability

Supporting raw data may be obtained through special request from the corresponding author.

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
