# Peer review of "Word Recognition with a Cochlear Implant in Relation to Prediction and Electrode Position"

_jcm, 2023, doi:10.3390/jcm13010183_

Round 1

Reviewer 1 Report

Comments and Suggestions for Authors

Congratulations to the authors for their work. The article is interesting and useful for predicting cochlear implant outcomes. It is well-designed and written, and the discussion covers interesting topics.

Three minor changes should be considered: 

Line 48: Consider changing "variability of" to "variability in."

Line 48: Consider changing "contact to the round window of depending on the electrode array" to "contact to the round window depending on the electrode array."

Line 147: It appears that "the CI522" is repeated.

Comments on the Quality of English Language

Three minor changes should be considered: 

Line 48: Consider changing "variability of" to "variability in."

Line 48: Consider changing "contact to the round window of depending on the electrode array" to "contact to the round window depending on the electrode array."

Line 147: It appears that "the CI522" is repeated.

Reviewer 2 Report

Comments and Suggestions for Authors

Dear authors. This is highly commendable manuscript and I thoroughly enjoyed reading it. I have just some suggestions and clarifications for you to consider. 

-----------------------------------------------------------------

Line 75 electrophysiological parameters, "scale">>> scalar 

Like 76-77 on "the" one hand, prediction models can contribute to more precise patient counselling. On the other, 

remove "the"

remove full stop after patient counseling and add the word "and" 

>>> and on the other,

Line 98-99

For hearing threshold beyond the maximum possible presentation levels of the audiometers, a value of 120 dbHL was "imputed>>>assigned"

assigned may be a better word 

Question: inclusion criteria is to include only patients with PBmax >0%. It is unclear if those with Audiogram at the limits and assigned 120db HL were excluded in this study as most likely they have cochlear dead regions and have 0% WRS max ? Unless those with 120dbHL audiogram all had >0% WRS max and were included.

If so, How was WRS max measured for those with Audiogram limits at 120db HL? 

Line 103 

The study was (also) registered.. 

add the word also

Line 108 

To identify the WRS max, the presentation level was increased in steps of 10dB until maximum score achievable… 

Please clarify and cite the reference that this is a valid method of seeking PB max as usually it is +30/40 sensation levels above the SRT. 

Line 116

To calculate the prognoses of the WRS(CI) with the cochlear implant equation was used 

Remove the word “with” and add a comma 

Lines 174-176

The positive and stronger correlation between angular insertion depth and d makes sense as the deeper the insertion, the further away the first basal electrode is from the Round window. 

Negative correlation is weak looking at the wider variance and spread in the plot and intuitive this makes sense too as we measure A (diameter) from a fixed point and how deep the angular depth shouldn’t affect fixed point measurements) 

Line 200-201

Please clarify as more than half your patients has WRS 65(HA) of 0%. If in aided condition it’s 0%, I assume WRS max unaided is also 0%. Your inclusion criteria seems to be for patients with >0% WRSmax?

Line 205 

These results suggest that especially patients with a preoperatively high difference between… 

Remove word especially and replace word high with great

Comments on the Quality of English Language

minor edits suggested.
